

# Establishment and verification of a nomogram model based on the inflammatory indicators of patients with cervical cancer for predicting the risk of their lymph node metastasis

Liyun Song, Kaiyun Qin, Suning Bai, Qi Wu, Jing Zhao, Jie Qi, Junmei Zhang, Yazhuo Wang, Yuan Zhang and Ren Xu

Department of Gynecology, Hebei General Hospital, Hebei Medical University, Shijiazhuang, Hebei, China

## ABSTRACT

Based on inflammatory indicators, this study aimed to predict the risk of lymph node metastasis (LNM) in patients with cervical squamous cell carcinoma (CSCC) and establish a predictive nomogram model.

**Methods**. This retrospective study analyzed the clinical data of 194 patients with stage IA2-IIA2 who underwent surgery at Hebei General Hospital (between January 2017 and August 2023). Patients were divided into two groups based on the presence of LNM or not. Clinical data of the participants were gathered and analyzed to compare the two groups. Logistic regression analysis was used to analyze the factors influencing LNM in patients with CSCC. R software was used to construct a nomogram model to predict LNM in patients with CSCC, and its accuracy was verified.

**Results**. Squamous cell carcinoma antigen (SCC-Ag) level, D-dimer level, platelet (PLT) count, and platelet-to-lymphocyte ratio (PLR) index were significantly higher in patients with LNM than in those without LNM ($P < 0.05$). There was a significant association between lymph vascular space invasion (LVSI) in patients with CSCC and their LNM ($P < 0.05$). Logistic regression analysis showed that SCC-Ag, PLR, and LVSI in patients with CSCC were independent risk factors for LNM ($P < 0.05$). A predictive nomogram model was constructed, and the prediction probability was consistent with the actual observed value (Hosmer–Lemeshow $P = 0.313$). Analyses using the receiver operating characteristic (ROC) curve revealed that the combination of the SCC-Ag, PLR, and LVSI values of patients with CSCC significantly improved the diagnostic efficiency of their LNM (AUC = 0.792, $P < 0.001$).

**Conclusion**. Establishing a nomogram model based on preoperative inflammatory indicators of patients with CSCC can accurately predict the risk of LNM, providing evidence for implementing a clinical diagnosis and treatment scheme.

Corresponding author
Liyun Song,
hehe271565227@126.com

## INTRODUCTION

Cervical cancer (CC) is one of the most common gynecological malignances in women worldwide and one of the major causes of cancer-related deaths in women. Approximately 604,000 new cases of CC are diagnosed every year, and CC causes approximately 341,000 deaths annually worldwide (*Sung et al., 2021*). The prognosis for those with metastatic CC is unfavorable (*Li, Wu & Cheng, 2016*). Similar to most other cancers, radical surgery is the recommended treatment for early-stage CC. Precise surgical staging after removal of the primary tumor lesion can assist in choosing the most suitable postoperative adjuvant therapy (*Levy et al., 2009*).

CC metastasis can be classified into direct extension, spread through the lymphatic system, and spread through the bloodstream. Lymph node metastasis (LNM) is one of the most important prognostic factors for CC. Although the vast majority of women with early-stage CC have a favorable prognosis, those with LNM are at a higher risk of relapse and mortality (*Huang & Fang, 2018*). LNM is associated with decreased overall survival and increased recurrence rate (*Matsuo et al., 2019*). Consequently, individuals with LNM need to undergo postoperative therapy, highlighting the significance of the precise assessment of LNM in designing personalized treatment strategies, enhancing the survival rate, and decreasing the mortality rate (*Carlson et al., 2016*). Lymphadenectomy is commonly used to assess lymph node status in early CC; however, there remains debate over its application principle (*Shah et al., 2011*).

Despite the potential benefits of pelvic lymphadenectomy in improving the long-term survival of CC patients with LNM, the decision to undergo an extensive lymphadenectomy may come with risks of unnecessary complications like blood vessel injuries, nerve injuries, infection, lymphocytes, and lymphedema (*Khatun et al., 2017*; *Obrzut et al., 2017*; *Barquet-Muñoz et al., 2017*). Lymphadenectomy can have a severe impact on patients' quality of life, leading to extended hospital stays, higher medical costs, and put lives at risk. However, only a small percentage of patients with early-stage CC have LNM. Incomplete removal of metastatic lymph nodes may occur with selective lymph node dissection, leading to a higher chance of recurrence and metastasis. Therefore, it is crucial to preoperatively identify patients with a low likelihood of LNM. It can help decrease the number of unnecessary radical lymphadenectomies in patients with early-stage disease.

Imaging technology is commonly utilized to evaluate LNM in CC patients. Nevertheless, preoperative tests such as endoscopic ultrasonography and computed tomography (CT) do not provide precise LNM assessments (*Togami et al., 2014*; *Martínez et al., 2013*; *Liu, Gao & Li, 2017*). Therefore, clinicians must make comprehensive judgments that consider the high-risk clinicopathological factors.

Tumor metastasis through lymphatic vessels typically begins at the sentinel lymph node (SLN), which is important for the diagnosis and treatment of diseases. Evidence suggests that SLN biopsy can be used for lymph node staging in patients with early CC. In a study by *Wydra et al. (2006)*, 100 patients with early CC were analyzed, showing detection rates using SLN as follows: 96.6% for Ib1 stage, 66.7% for Ib2 stage, and 62.5% for stage IIa. A false negative rate of 3% was observed. Nonetheless, the diagnostic reliability of sentinel

lymph node biopsy (SLNB) may be influenced by the tumor size, staging, and additional factors. Routine frozen section analysis is necessary to diagnos CC using intraoperatively resected SLNs. Nevertheless, multiple studies have shown that intraoperative frozen section analysis is unreliable for detecting micrometastases (*Huang & Fang, 2018*). Consequently, this technique has not been extensively validated (*Cibula & McCluggage, 2019*; *Cibula et al., 2020*; *Zhang et al., 2021*; *Lecuru et al., 2019*; *Bizzarri et al., 2021*).

Mounting evidence indicates a correlation between nutritional status, inflammation, and survival rates in different types of cancer. Many studies have demonstrated that various factors, such as nutritional markers and inflammation levels, can predict the prognosis of various types of tumors (*Kim et al., 2021*; *Bullock et al., 2020*; *Wang et al., 2021*; *Kumarasamy et al., 2019*). A series of systemic inflammatory indices, such as the systemic immune-inflammation index (SII), prognostic nutritional index (PNI), fibrinogen-to-albumin ratio (FAR), neutrophil-to-lymphocyte ratio (NLR), platelet-to-lymphocyte ratio (PLR), and monocyte-to-lymphocyte ratio (MLR), can be acquired easily and affordably. To our knowledge, there is a shortage of documented studies investigating the efficacy of the systemic inflammatory indices mentioned above in predicting the risk of LNM in CC before surgery.

Nomograms are prevalent prognostic tools in oncology and medicine, valued for generating individualized numerical probabilities of clinical events by combining multiple prognostic and determinant variables (*Balachandran et al., 2015*). This aligns with the goals of biologically/clinically integrated models and personalized medicine. Key advantages—rapid digital computation, increased accuracy, and more understandable prognoses than conventional staging—make nomogram-derived estimates easy to incorporate into clinical decision support (*Balachandran et al., 2015*).

The objective of this study was to identify independent predictors of inflammatory markers and to establish and validate a nomogram to predict the risk of LNM in patients with cervical squamous cell carcinoma (CSCC). Using the nomogram, patients at low risk for LNM can be identified, helping to determine the necessity for adjunctive therapy and predict prognosis.

## MATERIALS AND METHODS

### Inclusion and exclusion criteria

This study retrospectively included 194 patients with CSCC who underwent surgery at Hebei General Hospital between January 1, 2017, and August 1, 2023. The Ethics Committee of Hebei General Hospital approved this study in accordance with the Declaration of Helsinki. The time when data were accessed for research purposes was 10/5/2024. The inclusion criteria were as follows: (1) all necessary information regarding the patient's age, medical history, previous treatment, and other basic clinical details is complete; (2) pathologically confirmed primary SCC of the uterine cervix, a clinical diagnosis of International Federation of Gynecology and Obstetrics (FIGO) stage IA2-IIA2 disease; (3) all patients underwent radical hysterectomy with/without oophorectomy with pelvic lymphadenectomy with/without para-aortic lymphadenectomy. Exclusion criteria:

(1) other types of CC other than CSCC; (2) patients who had not undergone pelvic lymphadenectomy; (3) patients missing complete clinical and pathological records; (4) patients with tumors spread to the peritoneum or distant metastatic sites; (5) patients with other diagnoses of cancer; (6) patients with other benign disorders affecting serum SCC-Ag; (7) patients with infectious diseases, autoimmune conditions, severe liver or kidney issues, thrombotic disorders, pregnancy, or preexisting blood disorders.

## Clinical and laboratory data collection

Clinical information, LNM status, and preoperative hematological data were collected for all patients. Clinical information included patient age, FIGO stage, body mass index (BMI), hypertension, and diabetes mellitus. The pathological features mainly included the lymph vascular space invasion (LVSI) and LNM. One week before treatment, the following hematological parameters were obtained: albumin, fibrinogen, neutrophil count, monocyte count, platelet count, lymphocyte count, D-dimer level, and SCC-Ag level. The combination of ultrasound, MR, and CT scans, along with an enhancement scan, helps to assess local tumor infiltration, evaluate the spread of retroperitoneal lymph nodes, and detect distant metastases. Positron emission tomography/computed tomography (PET-CT) can be used to rule out metastatic diseases. Two senior pathologists reviewed the pathological examinations, and the Ethics Committee of Hebei General Hospital approved the collection of the patients' clinical and laboratory data, following the Declaration of Helsinki. The ethical committee's conclusion that informed consent was not necessary meant that written informed consent was no longer necessary. The patient information was de-identified. This retrospective study adhered to the STROBE guidelines (*Von Elm et al., 2007*).

## Inflammation-related markers

The following formula was used for serum inflammation-related markers: FAR = fibrinogen (g/L)/albumin (g/L); PNI = albumin (g/L) + 5 × lymphocyte count ($10^9$/L); SII = platelet count ($10^9$/L) × neutrophil count ($10^9$/L)/lymphocyte count ($10^9$/L); NLR = neutrophil count ($10^9$/L)/lymphocyte count ($10^9$/L); PLR = platelet count ($10^9$/L)/lymphocyte count ($10^9$/L); and MLR = monocyte count ($10^9$/L)/lymphocyte count ($10^9$/L).

## Statistical analysis

Data analysis was performed using IBM SPSS V26.0 (IBM Corp., Armonk, NY, USA), GraphPad Prism 9.0 (GraphPad Software, Inc., La Jolla, CA, USA), and the R Environment for Statistical Computing software. A $P$ value less than 0.05 indicated statistical significance. The normality of the variable distribution was evaluated using the Shapiro–Wilk test. For continuous variables following a normal distribution, the data are presented as the mean ± SD. The median and interquartile ranges are displayed for those that did not follow a normal distribution. The Kruskal–Wallis test was used to assess differences among groups, and multiple comparisons were conducted using the Mann–Whitney $U$ test. Receiver operator characteristic (ROC) analysis was employed to test the diagnostic performance of the identified variables in predicting CC, with calculations made for the area under the ROC curve (AUC), 95% confidence interval (CI), sensitivity, and specificity.

Key parameters were analyzed using univariate and multivariate logistic regression analyses to develop a nomogram that can predict pelvic LNM. Calibration curves were plotted and assessed based on the agreement between predictions and observations, and the Hosmer–Lemeshow goodness-of-fit test was performed. A bootstrap procedure with 10,000 resamples was conducted to internally validate the nomograms. The developed model was characterized based on area under the curve (AUC) and ROC.

## RESULTS

### Baseline information of SCC patients

The clinical data of 194 patients who had initially been diagnosed with CSCC and treated surgically in the hospital from January 1, 2017, to August 1, 2023, were included in this retrospective study. There were 15 patients with stage IA2, 52 with stage IB1, 50 with stage IB2, 22 with stage IB3, 34 with stage IIA1, and 21 with stage IIA2. Of the 194 patients with CSCC included in this study, 139 (71.65%) had FIGO stage I. These patients were divided into two groups based on LNM: the group with LNM confirmed by pathological examination (presence of LMN, $n = 37$) and the group without LNM (absence of LMN, $n = 157$). Clinical information of the female participants was gathered and analyzed to compare the two groups. Patient characteristics are presented in Table 1. Our study showed an LNM rate of 19.07% (37/194), largely consistent with existing research suggesting that 10–30% of early-stage CC patients develop LNM (*Huang & Fang, 2018*; *Kwon et al., 2018*). Of the 37 patients with LNM, 26 (70.3%) tested positive for LVSI.

The age range of the lymph-positive CSCC patients was 31–70 years, and that of the lymph-negative patients was 28–78 years. Age did not differ significantly between the two groups. Table 1 presents evidence of notable differences in LVSI, SCC-Ag, platelet (PLT), PLR, and D-dimer between lymph-positive and lymph-negative patients ($P < 0.001$, $P < 0.001$, $P = 0.01$, $P = 0.034$, $P < 0.001$, respectively). However, no significant differences were found in the absolute neutrophil count, absolute lymphocyte count, absolute monocyte count, SII, NLR, MLR, FAR, and PNI between the two groups ($P = 0.460$, $P = 0.703$, $P = 0.323$, $P = 0.056$, $P = 0.325$, $P = 0.325$, $P = 0.306$, $P = 0.924$, respectively). Moreover, there was no correlation between LNM and factors such as BMI, diabetes, and hypertension.

We created ROC curves by analyzing the connection between SCC-Ag, PLT, PLR, D-dimer, and LNM in CSCC. We determined the optimal cut-off value, sensitivity, and specificity (Fig. 1). Table 2 presents the main statistics. The optimum cutoff value was chosen to maximize the Youden index (sensitivity + specificity − 1). The appropriate cutoff values of SCC-Ag (AUC = 0.705, CI [0.635–0.769], $P < 0.001$), PLT (AUC = 0.636, CI [0.564–0.704], $P = 0.004$), PLR (AUC = 0.612, CI [0.539–0.681], $P = 0.039$), and D-dimer (AUC = 0.651, CI [0.579–0.718], $P = 0.001$) for differentiating lymph-positive and lymph-negative patients were 5.631, 207, 248.73, and 0.27, respectively, with corresponding sensitivities of 68.6%, 97.30%, 27.03%, and 86.11%, respectively, and specificities of 70.1%, 26.11%, 94.27%, and 46.50%, respectively.

**Table 1  Baseline characteristics of patients with SCC.**

| Variables | Lymph node positive ($n = 37$) | Lymph node negative ($n = 157$) | (Mann–Whitney U) $Z/x^2$ | *P*-value |
|---|---|---|---|---|
| Age (years, Median (IQR)) | 56 (15) | 53 (17) | −0.169 | 0.866 |
| BMI (kg/m$^2$, Median (IQR)) | 24.82 (5.28) | 25.07 (4.48) | −0.946 | 0.344 |
| Diabetes, n (%) | 4 (10.81) | 23 (14.65) | 0.368 | 0.544 |
| Hypertension, n (%) | 10 (27.03) | 40 (25.48) | 0.038 | 0.846 |
| LVSI, n (%) | 26 (70.27) | 57 (36.31) | 14.111 | <0.001 |
| SCC-Ag (ng/ml, Median (IQR)) | 8.78 (12.13) | 2.86 (5.75) | −3.793 | <0.001 |
| N ($10^9$/L, Median (IQR)) | 4.17 (2.26) | 3.96 (1.94) | −0.739 | 0.460 |
| L ($10^9$/L, Median (IQR)) | 1.70 (0.86) | 1.62 (0.66) | −0.381 | 0.703 |
| M ($10^9$/L, Median (IQR)) | 0.30 (0.14) | 0.27 (0.11) | −0.989 | 0.323 |
| PLT ($10^9$/L, Median (IQR)) | 274.00 (81.00) | 248.00 (84.00) | −2.570 | 0.010 |
| SII ($10^9$/L, Median (IQR)) | 638.24 (688.36) | 575.56 (417.47) | −1.912 | 0.056 |
| NLR, Median (IQR) | 2.47 (1.64) | 2.28 (1.52) | −0.985 | 0.325 |
| PLR, Median (IQR) | 165.95 (129.81) | 148.53 (74.93) | −2.116 | 0.034 |
| MLR, Median (IQR) | 2.47 (1.64) | 2.28 (1.52) | −0.985 | 0.325 |
| FAR, Median (IQR) | 0.07 (0.03) | 0.07 (0.02) | −1.024 | 0.306 |
| PNI, Median (IQR) | 51.98 (6.30) | 50.45 (5.50) | −0.096 | 0.924 |
| D-Dimer (mg/L) | 0.37 (0.31) | 0.29 (0.29) | −2.819 | <0.001 |

**Notes.**

N, absolute neutrophil count; L, absolute lymphocyte count; M, monocyte count; PLT, blood platelet count; SII, platelet count ($10^9$/L) × neutrophil count ($10^9$/L)/lymphocyte count ($10^9$/L); NLR, neutrophil count ($10^9$/L)/lymphocyte count ($10^9$/L); PLR, platelet count ($10^9$/L)/lymphocyte count ($10^9$/L); MLR, monocyte count ($10^9$/L)/lymphocyte count ($10^9$/L); FAR, fibrinogen(g/L)/albumin(g/L); PNI, albumin (g/L) + 5 × lymphocyte count ($10^9$/L).

## Univariate and multivariate predictors for LNM

Independent risk factors linked to LNM occurrence were determined using univariate and multivariate logistic regression analyses. In univariate analysis, positive LVSI, serum SCC-Ag ≥5.63 ng/ml, PLT≥207 × 109/L, and PLR ≥248.739 were associated with pelvic LNM in CSCC. Multivariate logistic analysis revealed that the following parameters were independent factors for pelvic LNM: LVSI (odds ratio (OR) = 6.182, 95% confidence interval (CI) [2.463–15.513], $P < 0.001$), serum SCC-Ag ((OR) = 1.047, 95% CI [1.018–1.078], $P = 0.001$), and PLR ((OR) = 1.008, 95% CI [1.001–1.014], $P = 0.021$). The univariate and multivariate logistic regression analyses are outlined in Table 3.

## Construction and validation of the nomogram model for predicting pelvic LNM

This study eliminated non-significant variables to create a high-performing association model with strong predictive capabilities. Using the results of the abovementioned multivariate analysis, a nomogram was formulated to estimate LNM, incorporating serum SCC-Ag, LVSI, and PLR. This nomogram enabled the evaluation of LNM risk for individual patients, highlighting the impact of each risk factor on the overall risk of lymph node metastasis (Fig. 2). Patient characteristics were placed on the variable line. A vertical line was drawn at the corresponding point on the point line to assign a point value to the

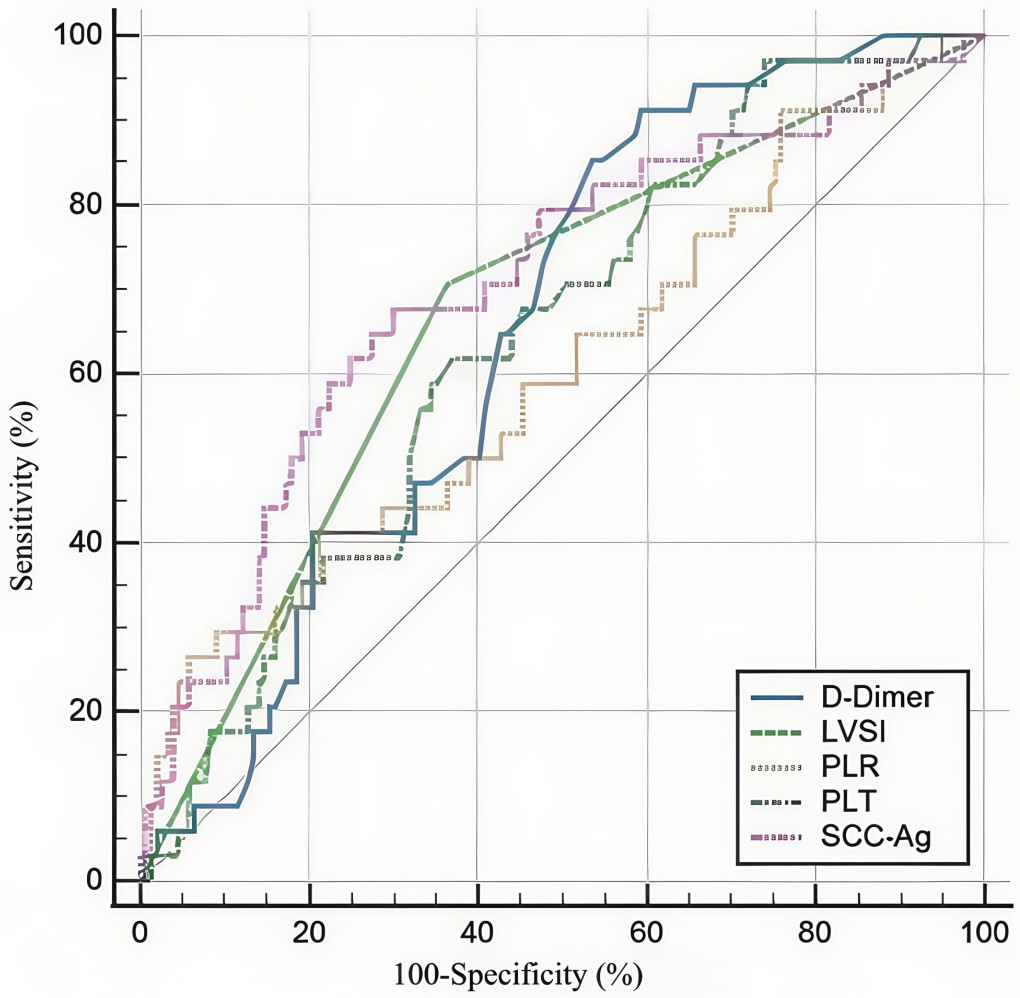

**Figure 1** **Receiver operating characteristic curves for the use of SCC-Ag, PLT, PLR, and D-Dimer to predict LNM of SCC patients in stage IA2-IIA2.** SCC-Ag, squamous cell carcinoma antigen; PLT, platelet; PLR, platelet-to-lymphocyte ratio; LNM, lymph node metastasis.

variable. The probability of LNM was calculated by summing all the points and drawing a vertical line over the total rows of points.

The concordance index (C-index) is a generalization of the area under the ROC curve. It quantifies the ability to distinguish low- and high-risk patients (an utterly random prediction had a C-index of 0.5, and a perfect rule had a C-index of 1.0). It was almost equal to the AUC of the ROC curve, which plotted the sensitivity against 1—specificity to predict the outcome (Fig. 3). In our study, the AUC for the nomogram constructed to predict LNM was 0.792 (95% CI [0.704–0.879]).

By comparing the nomogram with an ideal curve, we developed a calibration curve that displayed a high level of agreement between the predicted and observed results in the

**Table 2** Cut-off value and diagnostic value of SCC-Ag, PLT, PLR, and D-Dimer.

| Variables | AUC | Cut-off | 95%CI | P-value | Sensitivity (%) | Specificity (%) |
|---|---|---|---|---|---|---|
| SCC-Ag | 0.705 | 5.631 | 0.635–0.769 | <0.001 | 68.6 | 70.1 |
| PLT | 0.636 | 207 | 0.564- 0.704 | 0.004 | 97.30 | 26.11 |
| PLR | 0.612 | 248.739 | 0.539–0.681 | 0.039 | 27.03 | 94.27 |
| D-Dimer | 0.651 | 0.27 | 0.579–0.718 | 0.001 | 86.11 | 46.50 |

Notes.
PLT, blood platelet count; PLR, platelet count ($10^9$/L)/lymphocyte count ($10^9$/L); AUC, area under the curve; CI, confidence interval.

**Table 3** Univariate and multivariate logistic regression analyses of variables.

| Variable | Character | Univariate | | Multivariate | |
|---|---|---|---|---|---|
| | | OR (95% CI) | p-value | OR (95% CI) | p-value |
| LVSI | Absent | 1.0 (Reference) | <0.001 | Reference | <0.001 |
| | Present | 4.147 (1.908–9.013) | | 6.182 (2.463–15.513) | |
| SCC-Ag (ng/ml) | <5.631 | 1.0 (Reference) | 0.003 | Reference | 0.001 |
| | ≥5.631 | 1.039 (1.013–1.066) | | 1.047 (1.018–1.078) | |
| PLT ($10^9$/L) | <207 | 1.0 (Reference) | 0.013 | Reference | 0.271 |
| | ≥207 | 1.008 (1.002–1.014) | | 1.004 (0.997–1.012) | |
| PLR | <248.739 | 1.0 (Reference) | 0.005 | Reference | 0.021 |
| | ≥248.739 | 1.008 (1.002–1.013) | | 1.008 (1.001–1.014) | |
| D-Dimer (mg/L) | <0.27 | 1.0 (Reference) | 0.237 | | |
| | ≥0.27 | 1.108 (0.935–1.314) | | | |

Notes.
LVSI, lymph vascular space invasion; PLT, blood platelet count; PLR, platelet count ($10^9$/L)/lymphocyte count ($10^9$/L); CI, confidence interval.

calibration plot (Fig. 4). The Hosmer–Lemeshow test was employed to assess the fit of the model. The *p*-value was 0.313, demonstrating a good model fit.

# DISCUSSION

CC screening programs have contributed to worldwide growth in the early detection of CC (*Cohen et al., 2019*). The squamous cell type constitutes the majority (>85%) (*Sturgeon et al., 2010*). Lymphatic metastasis is prevalent in CSCC and serves as a key factor in determining the likelihood of recurrence and survival (*Nanthamongkolkul & Hanprasertpong, 2018*). Patients with pelvic LNM have a considerably lower 5-year survival rate than those without metastases (*Sakuragi, 2007*). Nevertheless, ongoing debate surrounds routine lymphadenectomy (*Aslan et al., 2020*; *Wang et al., 2020*). Not only does lymphadenectomy prolongs the operating time and increases blood loss, it also results in numerous serious complications. Nevertheless, LNM is present in only 10–30% of patients with early-stage CC (*Huang & Fang, 2018*; *Kwon et al., 2018*), and CSCC is less likely to progress to LNM than adenocarcinoma (*Xie et al., 2018*). This leads to concerns regarding whether patients with CC are undergoing unnecessary treatment. Therefore, it is essential
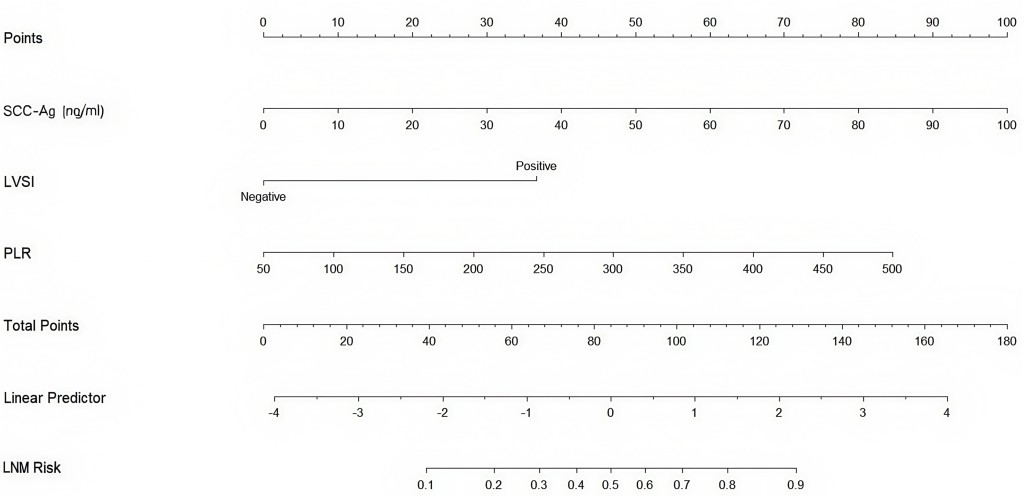

**Figure 2   Nomogram predicting the risk of lymph node metastasis in early-stage cervical cancer patients.** SCC-Ag, squamous cell carcinoma antigen; LVSI, lymph vascular space invasion; PLR, platelet-to-lymphocyte ratio; LNM Risk, lymph node metastasis risk.

to precisely evaluate patients at risk of LNM to develop individualized treatment regimens and improve prognosis.

In recent years, there has been a significant focus on the preoperative evaluation of LNM. Typically, CT or magnetic resonance imaging (MRI) is utilized to assess LNM in patients with CC, focusing primarily on lymph node size (*Lee & Atri, 2019*). However, these tools are not reliable for LNM assessment. PET/CT is more sensitive than CT or MRI alone, but it comes with a higher cost and limited accessibility (*Liu et al., 2021*). Therefore, numerous scientists have suggested the utilization of SLN biopsy to assess the condition of lymph nodes (*Bogani et al., 2024*). The procedure involves injecting a dye into the cervix, which travels to the sentinel nodes. Nevertheless, its regular use in clinical settings remains constrained by technical challenges such as intraoperative frozen sections and ultra-staging. Moreover, *Cibula & McCluggage (2019)* emphasized that this technique is unfeasible in clinical practice because the identified micrometastases (MIC) and isolated tumor cells (ITCs) require hundreds of slides per SLN. Therefore, it is evident that a certain proportion of MIC and ITCs are present in SLN-negative patients (*Cibula & McCluggage, 2019*). More studies are necessary to confirm the feasibility of SLN biopsy in cervical malignancy (*Cibula et al., 2020*; *Lecuru et al., 2019*).

Nomograms are a pictorial representation of a complex mathematical formula (*Grimes, 2008*). It has recently attracted the attention of many researchers. These graphical charts, utilizing logistic or Cox regression, offer an intuitive and simple way to predict event probabilities accurately. Nomograms are valuable for malignant tumors because they allow the risk to be evaluated using noninvasive or minimally invasive methods prior to radical surgery. Using a nomogram for individualized prediction could aid decision-making for both surgeons and patients. *Kim et al. (2014)* formulated a reliable model that utilized preoperative parameters to estimate the likelihood of LNM in patients with early CC.

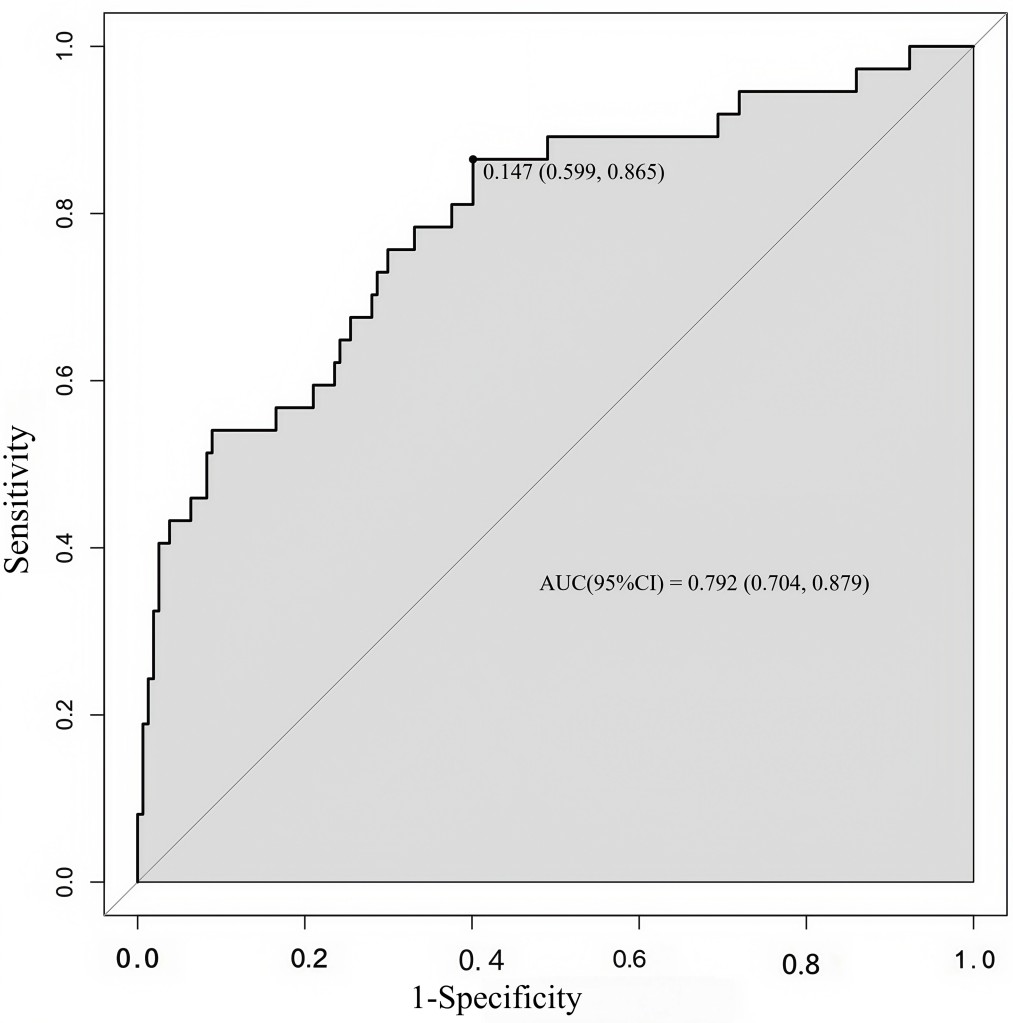

**Figure 3** **ROC curve of the developed model.** ROC, operating characteristic curve; AUC, greater area under the ROC curve.

The model effectively distinguished patients with a low LNM risk. Nomograms utilizing clinicopathological parameters show a strong predictive capability for assessing the risk of LNM and compensate for the limitations of imaging accuracy.

Inflammation is a vital characteristic of the tumor microenvironment and plays a central role in tumor initiation, promotion, progression, invasion, and metastasis (*Grivennikov, Greten & Karin, 2010*). The concept of the tumor microenvironment is built upon the "seed-soil" theory, initially proposed by British surgeon Stephen Paget. Both tumor and stromal cells produce numerous inflammatory factors, resulting in a complex inflammatory microenvironment within the cellular network. This inflammatory microenvironment plays a crucial role in affecting the malignant characteristics of tumors through the regulation of the biological processes involved in their development (*Arneth, 2019*; *Kulbe et al., 2012*). The immune and nutritional conditions of the body strongly influence the

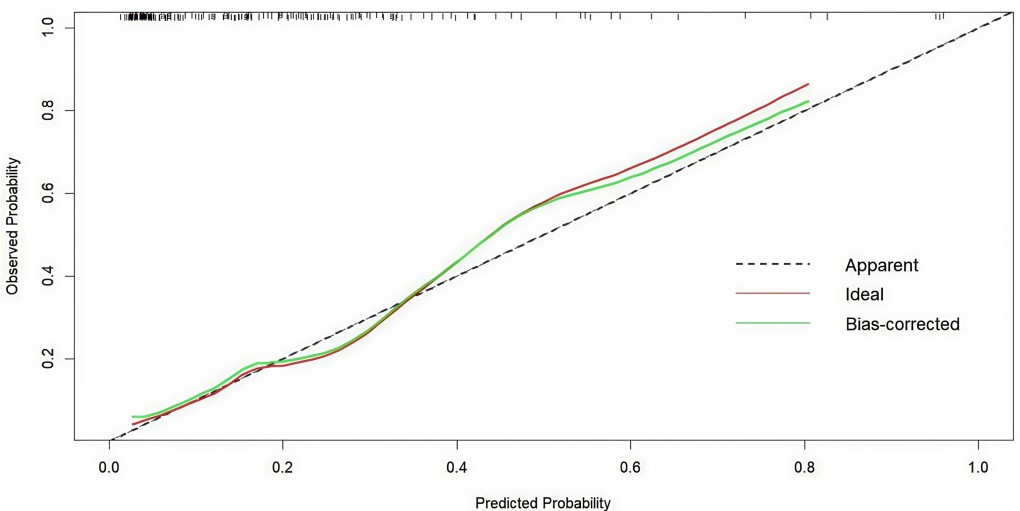

**Figure 4** **Internal verification plots of the nomogram calibration curves by 10-fold cross-validation.**

inflammatory response. Cancer progression can induce systemic inflammation, which is a crucial contributor to malnutrition and weakened immunity, both of which can lead to a substantial decrease in survival. Previous research suggests that malnutrition is associated with an increased likelihood of postoperative complications, heightened vulnerability to infections, and a possible rise in tumor recurrence as a result of compromised tumor immunity (*Sasahara et al., 2020*; *Yoshida et al., 2019*; *Ryan et al., 2009*).

Recent research has revealed that various factors, such as nutritional and inflammatory markers, can predict the outcome of diverse types of cancer. In this study, we aimed to identify independent prediction parameters from inflammatory parameters and to establish and validate a nomogram to predict the risk of lymph node metastasis in patients with cervical cancer. Utilizing the nomogram makes it possible to identify patients who are unlikely to have LNM, thus aiding in decisions regarding the need for additional therapy and prognosis determination.

Our study examined the diagnostic utility of albumin, fibrinogen, neutrophil count, monocyte count, platelet count, lymphocyte count, D-dimer, SCC-Ag, and six inflammatory-nutritional markers (PNI, NLR, PLR, MLR, SII, and FAR) in patients with early-stage SCC with LNM. The values of SCC-Ag, PLT, PLR, and D-dimer were significantly higher in patients with LNM than in those without LNM ($P < 0.001$, $P < 0.001$, $P = 0.01$, $P = 0.034$, $P < 0.001$, respectively). LVSI in patients with SCC was associated with LNM ($P < 0.001$).

SCC-Ag acts as an indicator of CC and plays a role in predicting and determining the prognosis of CSCC (*Kato & Torigoe, 1977*). SCC-Ag serves as a biomarker not only for diagnosing and predicting CSCC but also for identifying uremia, azotemia, diabetic nephropathy, and nephrotic syndrome. Diseases known to increase serum SCC-Ag levels were deliberately excluded from the cases examined in this study. *Zhu et al. (2021)*

identified SCC-Ag levels ≥2.75 ng/ml were identified as significant independent factors influencing LNM. Preoperative evaluations may show a high risk of pelvic LNM when SCC-Ag levels are >eight ng/ml, according to *Lin et al. (2000)*. Similarly, our research demonstrated that SCC-Ag levels >5.63 ng/ml independently increased the risk of LNM. Thrombocytosis, or abnormally elevated platelet counts in the blood, has been shown to be an important factor in predicting outcomes in patients with CC (*Cheng et al., 2017*). Moreover, platelets promote tumor growth, facilitate tumor cell extravasation, and aid tumor metastasis (*Haemmerle et al., 2018*). PLR was shown to have significant predictive value for identifying patients with tumor stages IIB and above, as well as LNM, according to *Wang et al.*'s investigation. According to their research, elevated PLR values correlate with gradual increases in platelet counts relative to lymphocytes across all tumor stages (*Wang et al., 2017*). Changes in PLR reflect the balance between host platelet-dependent inflammatory responses and lymphocyte-mediated anti-tumor immune responses. Platelets are involved in the progression and spread of tumors, whereas circulating lymphocytes are essential for inhibiting the growth and spread of tumor cells (*Ding et al., 2010*). D-dimer is the product of fibrinolysis. Elevated plasma D-dimer levels in gynecological cancers result from hyperactivation of both coagulation and fibrinolysis systems (*Ye et al., 2015*). Activation of the clotting-fibrinolytic system by tumor cells leads to the release of various fibrinolytic and hemostatic markers. This, in turn, leads to the stimulation of vascular endothelial cell growth and the promotion of new blood vessel formation, which is vital for tumor progression (*Luo et al., 2015*). This indicates that thrombotic markers, such as plasma D-dimer, could serve as potential indicators of occult malignancies. The current study's observation of markedly higher median plasma D-dimer levels in women with CC than in cancer-free women corroborates this important theory. A survey by *Peelay et al. (2023)* demonstrated notable connections between increased plasma D-dimer levels and CC along with disease stage and grade.

Univariate and multivariate logistic regression analyses were conducted to examine how clinical and pathological features are related to the occurrence of LNM. Univariate analysis indicated that positive LVSI, serum SCC-Ag ≥ 5.63 ng/ml, PLT level ≥ 207×109/L, and PLR ≥ 248.739 were notably linked to LNM. However, D-dimer did not show significance as a predictor in the univariate model. In a multivariate model, positive LVSI, serum SCC-Ag ≥5.63 ng/ml, and PLR ≥248.739 were viable predictors of LNM. The results of the univariate and multivariate logistic regression analyses are presented in Table 3. Factors without significance or clinical relevance were removed to obtain a predictive model with high accuracy. A nomogram was developed using the final multivariate model and clinical practice data to measure the risk of LNM (Fig. 2). The internal validation of the nomogram was performed using the jackknife cross-validation test. Furthermore, each patient's likelihood of LNM was estimated using a nomogram created from the remaining data, and the ability of the model to distinguish between patients with positive and negative LNM was assessed using the C-index. In the nomogram, positive LVSI, serum SCC-Ag ≥5.63 ng/ml, and PLR ≥248.739 were considered variables for predicting risk, with individualized points allocated to each. The predicted probability of LNM is equal to the sum of all points. The probability of LNM was computed by drawing a vertical line for every variable point

on the axis. Summing up the scores of all the variables resulted in a total score. A vertical line was drawn from the total score line down to the predicted probability bottom scale to determine the individual probability of LNM. In analyzing various factors, LVSI emerged as the most powerful indicator of LNM, with an OR of 6.182.

Our study showed that a nomogram is a valuable tool for predicting the risk of LNM, demonstrating its practicality in clinical settings for LNM prediction. However, it is essential to acknowledge the limitations of the present study. Single-center retrospective studies could have patient selection bias as a drawback. Moreover, because we have information from only one center, our verification process is limited to internal verification.

## CONCLUSION

In conclusion, we developed a nomogram based on three parameters, LVSI, serum SCC-Ag, and PLR, and achieved good results in predicting LNM in cervical squamous cell carcinoma.

**Abbreviations**

| | |
|---|---|
| **SII** | systemic immune-inflammation index |
| **FAR** | fibrinogen-to-albumin ratio |
| **PNI** | prognostic nutritional index |
| **NLR** | neutrophil-to-lymphocyte ratio |
| **PLR** | platelet-to-lymphocyte ratio |
| **MLR** | monocyte-to-lymphocyte ratio |
| **ROC** | operating characteristic curve |
| **AUC** | greater area under the ROC curve |
| **CSCC** | cervical squamous cell carcinoma |
| **SCC-Ag** | squamous cell carcinoma antigen |
| **PLT** | platelet |
| **LVSI** | lymph vascular space invasion |
| **LNM** | lymph node metastasis |
| **CC** | cervical cancer |
| **CT** | computed tomography |
| **SLN** | sentinel lymph node |
| **SLNB** | sentinel lymph node biopsy |
| **FIGO** | Federation of Gynecology and Obstetrics |
| **CI** | confidence interval |
| **OR** | odds ratio |
| **C-index** | concordance index |
| **CSCC** | cervical squamous cell carcinoma |
| **PET/CT** | Positron emission tomography/computed tomography |
| **MIC** | micro-metastases |
| **ITC** | isolated tumor cell |

## ACKNOWLEDGEMENTS

We would like to thank all doctors, nurses, patients, and their family members for their kindness in supporting this study.

### Funding
The authors received no funding for this work.

### Competing Interests
The authors declare there are no competing interests.

### Author Contributions

- Liyun Song conceived and designed the experiments, performed the experiments, analyzed the data, prepared figures and/or tables, authored or reviewed drafts of the article, and approved the final draft.
- Kaiyun Qin analyzed the data, authored or reviewed drafts of the article, and approved the final draft.
- Suning Bai conceived and designed the experiments, analyzed the data, prepared figures and/or tables, authored or reviewed drafts of the article, and approved the final draft.
- Qi Wu conceived and designed the experiments, analyzed the data, prepared figures and/or tables, authored or reviewed drafts of the article, and approved the final draft.
- Jing Zhao conceived and designed the experiments, prepared figures and/or tables, authored or reviewed drafts of the article, and approved the final draft.
- Jie Qi conceived and designed the experiments, performed the experiments, prepared figures and/or tables, authored or reviewed drafts of the article, and approved the final draft.
- Junmei Zhang conceived and designed the experiments, performed the experiments, prepared figures and/or tables, authored or reviewed drafts of the article, and approved the final draft.
- Yazhuo Wang analyzed the data, authored or reviewed drafts of the article, and approved the final draft.
- Yuan Zhang conceived and designed the experiments, prepared figures and/or tables, authored or reviewed drafts of the article, and approved the final draft.
- Ren Xu conceived and designed the experiments, authored or reviewed drafts of the article, and approved the final draft.

### Ethics
The following information was supplied relating to ethical approvals (i.e., approving body and any reference numbers):

This study was approved by the Ethics Committee of Hebei General Hospital. Research Ethics Review No. 2024-LW-074.

### Data Availability
Data is available in the Supplementary File.

## Supplemental Information

Supplemental information for this article can be found online at http://dx.doi.org/10.7717/peerj.20069#supplemental-information.

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
