# Peer review of "Establishment and verification of a nomogram model based on the inflammatory indicators of patients with cervical cancer for predicting the risk of their lymph node metastasis"

_PeerJ, doi:10.7717/peerj.20069_

## Round 0.1 · original submission · Major Revisions

**Language Note:** The review process has identified that the English language must be improved. PeerJ can provide language editing services - please contact us at [email protected] for pricing (be sure to provide your manuscript number and title). Alternatively, you should make your own arrangements to improve the language quality and provide details in your response letter. – PeerJ Staff

Reviewer 1 ·

Basic reporting

Language and Writing: The manuscript contains multiple grammatical and language issues, especially in the abstract, methods, results, and discussion sections, which significantly impair readability. The authors are strongly encouraged to seek professional English editing services.
References: Although the number of references is adequate, many of them are not closely aligned with the main content, resulting in a sense of redundancy. Additionally, key components—such as the rationale behind the construction of the nomogram—lack citation of authoritative sources.
Figures and Tables: The quality of figures and tables is suboptimal. Many are overly simplistic with unclear labels and incomplete legends, which diminishes their interpretability.
Manuscript Structure: The overall structure is somewhat disorganized. There is substantial redundancy between the results and discussion sections, and some paragraphs are overly lengthy with unclear focal points.

Experimental design

Study Design: This is a single-center, retrospective study lacking external validation. The manuscript does not provide a clear sample source description or a flow diagram of patient inclusion, raising concerns about selection bias.
Variable Selection: The process for selecting variables is unclear. The authors do not address potential multicollinearity, and the inclusion criteria for variables in the regression model are not well justified, which undermines the robustness of the nomogram.
Sample Size: Only 37 patients had LNM, making the multivariate logistic regression and model development underpowered and prone to overfitting.
Statistical Methods: Although the ROC analysis reports AUC values, confidence intervals for the curves are not provided, limiting the interpretability of the model’s performance.

Validity of the findings

Model Reliability: Due to limited sample size, unclear variable selection criteria, and lack of external validation, the proposed nomogram lacks robustness and generalizability.
Overstated Conclusions: The manuscript claims that the model can provide clinical guidance, but this is not substantiated by real-world validation or clinical decision analysis.
Missing Confounding Factors: Important pathological and clinical variables—such as tumor size, depth of invasion, and preoperative imaging findings—were not included in the analysis, which may significantly affect model accuracy.

Additional comments

Although the study’s intention—to explore noninvasive tools for reducing unnecessary lymphadenectomy—is commendable, the current manuscript contains several critical scientific flaws that prevent it from meeting the standards of high-quality academic publishing. The authors may consider substantial revisions in the following areas :
Include external or prospective multicenter validation;
Clarify statistical modeling protocols and variable selection criteria;
Improve language clarity and logical consistency;
Incorporate key pathological and clinical covariates.

Reviewer 2 ·

Basic reporting

.

Experimental design

see the section " basic reporting"

Validity of the findings

see the section " basic reporting"

Additional comments

I read with great interest the Manuscript titled " Establishment and verification of a nomogram model based on the inflammatory indicators of patients with cervical cancer for predicting the risk of their lymph node metastasis "
the topic is interesting enough to attract the readers’ attention.
Although the manuscript can be considered already of good quality, I would suggest following recommendations:
- Nodal status is one of the most important prognostic factors for patients with cervical cancer. The authors could extend the discussion by evaluating and citing current evidence about sentinel node mapping and prognostic outcomes related. I would be glad if the authors discuss this important point, referring to PMID: 38901291.
- The management of cervical disease has advanced significantly over recent decades, I suggest considering the improvements in factors crucial in the development, in early diagnosis, surgical techniques, adjuvant therapies and the safety of neoadjuvant chemotherapy and conization in early-stage cervical cancer. I suggest adding recent evidence about this topic: 39955179
Because of these reasons, the article should be revised and completed. Considering all these points, I think it could be of interest to the readers and, in my opinion, it deserves the priority to be published after minor revisions.

---

## Round 0.2 · accepted · Accept

Reviewer 1 did not agree to re-review, but I confirm that the authors have addressed all of the reviewers' comments and the manuscript is ready for publication.

Reviewer 2 ·

Basic reporting

The quality of the manuscript has improved thanks to the changes made. I think it could be of interest to the readers and, in my opinion, it deserves priority to be published.

Experimental design

-

Validity of the findings

-